# CROSS-DOMAIN FEW-SHOT LEARNING BY REPRESENTATION FUSION

## ABSTRACT

In order to quickly adapt to new data, few-shot learning aims at learning from few examples, often by using already acquired knowledge. The new data often differs from the previously seen data due to a domain shift, that is, a change of the input-target distribution. While several methods perform well on small domain shifts like new target classes with similar inputs, larger domain shifts are still challenging. Large domain shifts may result in high-level concepts that are not shared between the original and the new domain. However, low-level concepts like edges in images might still be shared and useful. For cross-domain few-shot learning, we suggest representation fusion to unify different abstraction levels of a deep neural network into one representation. We propose Cross-domain Hebbian Ensemble Few-shot learning (CHEF), which achieves representation fusion by an ensemble of Hebbian learners acting on different layers of a deep neural network that was trained on the original domain. On the few-shot datasets *mini*Imagenet and *tiered*Imagenet, where the domain shift is small, CHEF is competitive with state-of-the-art methods. On cross-domain few-shot benchmark challenges with larger domain shifts, CHEF establishes novel state-of-the-art results in all categories. We further apply CHEF on a real-world cross-domain application in drug discovery. We consider a domain shift from bioactive molecules to environmental chemicals and drugs with twelve associated toxicity prediction tasks. On these tasks, that are highly relevant for computational drug discovery, CHEF significantly outperforms all its competitors.

## 1 INTRODUCTION

Currently, deep learning is criticized because it is data hungry, has limited capacity for transfer, insufficiently integrates prior knowledge, and presumes a largely stable world (Marcus, 2018). In particular, these problems appear after a domain shift, that is, a change of the input-target distribution. A domain shift forces deep learning models to adapt. The goal is to exploit models that were trained on the typically rich original data for solving tasks from the new domain with much less data. Examples for domain shifts are new users or customers, new products and product lines, new diseases (e.g. adapting from SARS to COVID19), new images from another field (e.g. from cats to dogs or from cats to bicycles), new social behaviors after societal change (e.g. introduction of cell phones, pandemic), self-driving cars in new cities or countries (e.g. from European countries to Arabic countries), and robot manipulation of new objects.

Domain shifts are often tackled by meta-learning (Schmidhuber, 1987; Bengio et al., 1990; Hochreiter et al., 2001), since it exploits already acquired knowledge to adapt to new data. One prominent application of meta-learning dealing with domain shifts is few-shot learning, since, typically, from the new domain much less data is available than from the original domain. Meta-learning methods perform well on small domain shifts like new target classes with similar inputs. However, larger domain shifts are still challenging for current approaches. Large domain shifts lead to inputs, which are considerably different from the original inputs and possess different high-level concepts. Nonetheless, low-level concepts are often still shared between the inputs of the original domain and the inputs of the new domain. For images, such shared low-level concepts can be edges, textures, small shapes, etc. One way of obtaining low level concepts is to train a new deep learning model from scratch, where the new data is merged with the original data. However, although models of the original domain are often available, the original data, which the models were trained on, often are not. This might have several reasons, e.g. the data owner does no longer grant access to the data, General

Data Protection Regulation (GDPR) does no longer allow access to the data, IP restrictions prevent access to the data, sensitive data items must not be touched anymore (e.g. phase III drug candidates), or data is difficult to extract again. We therefore suggest to effectively exploit original data models directly by accessing not only high level but also low level abstractions. In this context, we propose a cross-domain few-shot learning method extracting information from different levels of abstraction in a deep neural network.

**Representation fusion.** Deep Learning constructs neural network models that represent the data at multiple levels of abstraction (LeCun et al., 2015). We introduce *representation fusion*, which is the concept of unifying and merging information from different levels of abstraction. Representation fusion uses a fast and adaptive system for detecting relevant information at different abstraction levels of a deep neural network, which we will show allows solving versatile and complex cross-domain tasks.

**CHEF.** We propose cross-domain ensemble few-shot learning (CHEF) that achieves representation fusion by an ensemble of Hebbian learners, which are built upon a trained network. CHEF naturally addresses the problem of domain shifts which occur in a wide range of real-world applications. Furthermore, since CHEF only builds on representation fusion, it can adapt to new characteristics of tasks like unbalanced data sets, classes with few examples, change of the measurement method, new measurements in unseen ranges, new kind of labeling errors, and more. The usage of simple Hebbian learners allows the application of CHEF without needing to backpropagate information through the backbone network.

The main contributions of this paper are:

- We introduce representation fusion as the concept of unifying and merging information from different layers of abstraction.

- We introduce CHEF[1] as our new cross-domain few-shot learning method that builds on representation fusion. We show that using different layers of abstraction allows one to successfully tackle various few-shot learning tasks across a wide range of different domains. CHEF does not need to backpropagate information through the backbone network.

- We apply CHEF to various cross-domain few-shot tasks and obtain several state-of-the-art results. We further apply CHEF to cross-domain real-world applications from drug discovery, where we outperform all competitors.

**Related work.** Representation fusion builds on learning a meaningful representation (Bengio et al., 2013; Girshick et al., 2014) at multiple levels of abstraction (LeCun et al., 2015; Schmidhuber, 2015). The concept of using representations from different layers of abstraction has been used in CNN architectures (LeCun et al., 1998) such as Huang et al. (2017); Rumetshofer et al. (2018); Hofmarcher et al. (2019), in CNNs for semantic segmentation in the form of multi-scale context pooling (Yu & Koltun, 2015; Chen et al., 2018), and in the form of context capturing and symmetric upsampling (Ronneberger et al., 2015). Learning representations from different domains has been explored by Federici et al. (2020); Tschannen et al. (2020) under the viewpoint of mutual information optimization. Work on domain shifts discusses the problem that new inputs are considerably different from the original inputs (Kouw & Loog, 2019; Wouter, 2018; Webb et al., 2018; Gama et al., 2014; Widmer & Kubat, 1996). Domain adaptation (Pan & Yang, 2009; Ben-David et al., 2010) overcomes this problem by e.g. reweighting the original samples (Jiayuan et al., 2007), learning features that are invariant to a domain shift (Ganin et al., 2016; Xu et al., 2019) or learning a classifier in the new domain. Domain adaptation where only few data is available in the new domain (Ben-David et al., 2010; Lu et al., 2020) is called cross-domain few-shot learning (Guo et al., 2019; Lu et al., 2020; Tseng et al., 2020), which is an instance of the general few-shot learning setting (Fei-Fei et al., 2006). Few-shot learning can be roughly divided into three approaches (Lu et al., 2020; Hospedales et al., 2020): (i) augmentation, (ii) metric learning, and (iii) meta-learning. For (i), where the idea is to learn an augmentation to produce more than the few samples available, supervised (Dixit et al., 2017; Kwitt et al., 2016) and unsupervised (Hariharan & Girshick, 2017; Pahde et al., 2019; Gao et al., 2018) methods are considered. For (ii), approaches aim to learn a pairwise similarity metric under which similar samples obtain high similarity scores (Koch et al., 2015; Ye & Guo, 2018; Hertz et al., 2006). For (iii), methods comprise *embedding and nearest-neighbor* approaches (Snell et al., 2017b;

---

[1]Our implementation is available at github.com/tomte812/chef.

Sung et al., 2018; Vinyals et al., 2016), *finetuning* approaches (Finn et al., 2017; Rajeswaran et al., 2019; Ravi & Larochelle, 2017; Andrychowicz et al., 2016), and *parametrized* approaches (Gidaris & Komodakis, 2018; Ye et al., 2020; Lee et al., 2019; Yoon et al., 2019; Mishra et al., 2018; Hou et al., 2019; Rusu et al., 2018). Few-shot classification under domain shifts for metric-based methods has been discussed in Tseng et al. (2020). Ensemble methods for few-shot learning have been applied in Dvornik et al. (2019), where an ensemble of distance-based classifiers is designed from different networks. In contrast, our method builds an ensemble of different layers from the same network. Hebbian learning as part of a few-shot learning method has been implemented in Munkhdalai & Trischler (2018), where fast weights that are used for binding labels to representations are generated by a Hebbian learning rule.

## 2 Cross-domain few-shot learning

**Domain shifts.** We assume to have data $(\boldsymbol{x}, \boldsymbol{y})$, where $\boldsymbol{x} \in \boldsymbol{X}$ is the input data and $\boldsymbol{y} \in \boldsymbol{Y}$ is the target data. A domain is a distribution $p$ over $\boldsymbol{X} \times \boldsymbol{Y}$ assigning each pair $(\boldsymbol{x}, \boldsymbol{y})$ a probability $p(\boldsymbol{x}, \boldsymbol{y})$. A domain shift is a change from $p(\boldsymbol{x}, \boldsymbol{y})$ to $\tilde{p}(\boldsymbol{x}, \boldsymbol{y})$. We measure the magnitude of the domain shift by a distance $d(p, \tilde{p})$ between the distributions $p$ and $\tilde{p}$. We consider four types of domain shifts (Kouw & Loog, 2019; Wouter, 2018; Webb et al., 2018; Gama et al., 2014; Widmer & Kubat, 1996):

- **Prior shift (small domain shift)**: $p(\boldsymbol{y})$ is changed to $\tilde{p}(\boldsymbol{y})$, while $p(\boldsymbol{x} \mid \boldsymbol{y})$ stays the same. For example, when new classes are considered (typical case in few-shot learning): $p(\boldsymbol{x}, \boldsymbol{y}) = p(\boldsymbol{y})p(\boldsymbol{x} \mid \boldsymbol{y})$ and $\tilde{p}(\boldsymbol{x}, \boldsymbol{y}) = \tilde{p}(\boldsymbol{y})p(\boldsymbol{x} \mid \boldsymbol{y})$.

- **Covariate shift (large domain shift)**: $p(\boldsymbol{x})$ is changed to $\tilde{p}(\boldsymbol{x})$, while $p(\boldsymbol{y} \mid \boldsymbol{x})$ stays the same. For example, when new inputs are considered, which occurs when going from color to grayscale images, using a new measurement device, or looking at traffic data from different continents: $p(\boldsymbol{x}, \boldsymbol{y}) = p(\boldsymbol{x})p(\boldsymbol{y} \mid \boldsymbol{x})$ and $\tilde{p}(\boldsymbol{x}, \boldsymbol{y}) = \tilde{p}(\boldsymbol{x})p(\boldsymbol{y} \mid \boldsymbol{x})$.

- **Concept shift**: $p(\boldsymbol{y} \mid \boldsymbol{x})$ is changed to $\tilde{p}(\boldsymbol{y} \mid \boldsymbol{x})$, while $p(\boldsymbol{x})$ stays the same. For example, when including new aspects changes the decision boundaries: $p(\boldsymbol{x}, \boldsymbol{y}) = p(\boldsymbol{x})p(\boldsymbol{y} \mid \boldsymbol{x})$ and $\tilde{p}(\boldsymbol{x}, \boldsymbol{y}) = p(\boldsymbol{x})\tilde{p}(\boldsymbol{y} \mid \boldsymbol{x})$.

- **General domain shift**: domain shift between $p(\boldsymbol{x}, \boldsymbol{y})$ to $\tilde{p}(\boldsymbol{x}, \boldsymbol{y})$. For example, going from Imagenet data to grayscale X-ray images (typical case in cross-domain datasets).

**Domain shift for images.** We consider the special case that the input $\boldsymbol{x}$ is an image. In general, domain shifts can be measured on the raw image distributions e.g. by using the $\mathcal{H}$-divergence (Ben-David et al., 2010). However, distances between raw image distributions were shown to be less meaningful in computer vision tasks than abstract representations of deep neural networks (Heusel et al., 2017; Salimans et al., 2016). We approximate the distance between the joint distributions $d(p(\boldsymbol{x}, \boldsymbol{y}), \tilde{p}(\boldsymbol{x}, \boldsymbol{y}))$ by the distance between the marginals $d(p(\boldsymbol{x}), \tilde{p}(\boldsymbol{x}))$, which is exact in the case of the covariate shift for certain choices of $d(\cdot, \cdot)$, like e.g. the Jensen-Shannon divergence. To measure the distance between the marginals $d(p(\boldsymbol{x}), \tilde{p}(\boldsymbol{x}))$ we use the Fréchet Inception Distance (FID; Heusel et al., 2017), i.e. the Wasserstein-2 distance of the features of the respective images activating an Inception v3 network (Szegedy et al., 2016) under a Gaussian assumption. The FID has proven reliable for measuring performance of Generative Adversarial Networks (Goodfellow et al., 2014).

**Cross-domain few-shot learning.** Large domain shifts lead to inputs, which are considerably different from the original inputs. As a result, the model trained on the original domain will not work anymore on the new domain. To overcome this problem, domain adaptation techniques are applied (Pan & Yang, 2009; Ben-David et al., 2010). Domain adaption can be achieved in several ways, e.g. by reweighting the original samples (Jiayuan et al., 2007). Another possibility is to learn a classifier in the new domain. Domain adaptation where in the new domain only few data is available (Ben-David et al., 2010) which can be used for learning is called cross-domain few-shot learning (Guo et al., 2019; Lu et al., 2020; Tseng et al., 2020). In an $N$-shot $K$-way few-shot learning setting, the training set (in meta learning also called one episode) consists of $N$ samples for each of the $K$ classes.

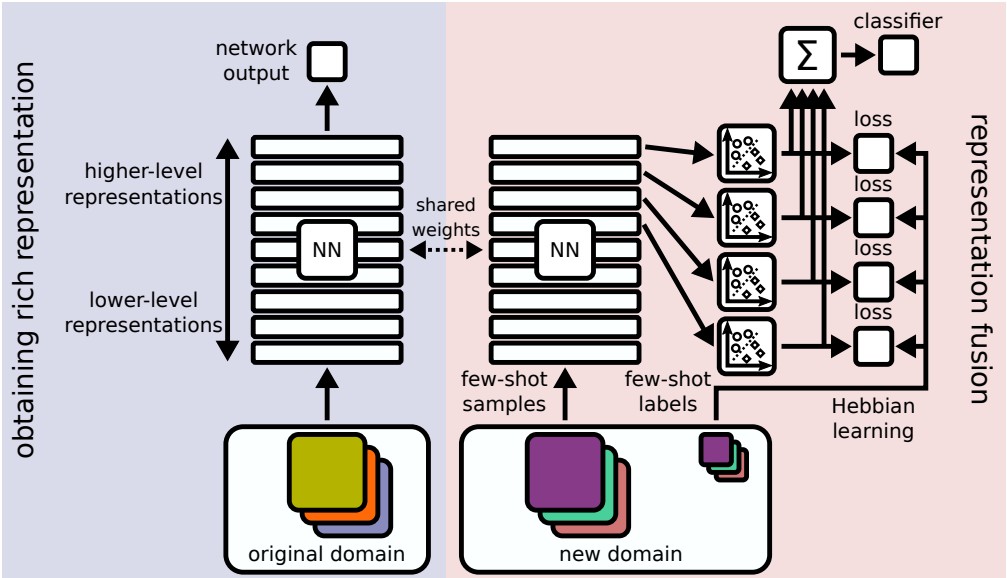

Figure 1: Working principle of CHEF. An ensemble of Hebbian learners is applied to the upper layers of a trained neural network. Distilling information from different layers of abstraction is called representation fusion. Each Hebbian learner is iteratively optimized and the results are combined.

## 3 CROSS-DOMAIN HEBBIAN ENSEMBLE FEW-SHOT LEARNING (CHEF)

We propose a new cross-domain few-shot learning method, CHEF, that consists of an ensemble of Hebbian learners built on representation fusion. Figure 1 sketches our CHEF approach. In principle, any learning algorithm can be used for representation fusion. We choose a Hebbian learning rule because it is simple and fast while being robust and reliable.

**Hebbian few-shot learning built on representation fusion.** CHEF builds its ensemble of Hebbian learners using representation fusion. Deep learning models (LeCun et al., 2015) provide hierarchical representations that allow to fuse information from different layers of abstraction. In contrast to many other methods, CHEF does not require backpropagation of error signals through the entire backbone network. Only the parameters of the Hebbian learners that are obtained for the uppermost layers need adjustment. This makes CHEF extremely fast and versatile.

**Obtaining one Hebbian Learner.** We consider an $N$-shot $K$-way few-shot learning setting. Let $z_i \in \mathbb{R}^D$ be a feature vector obtained from activating a pre-trained backbone network with a sample $x_i$ up to a certain layer, where $D$ is the number of units in that layer. We combine the $NK$ feature vectors into a matrix $Z \in \mathbb{R}^{NK \times D}$ and initialize a weight matrix $W \in \mathbb{R}^{K \times D}$. In accordance with Hebb (2005); Frégnac (2002), we use the Hebbian learning rule

$$W \leftarrow W - \alpha V^\top Z \tag{1}$$

for a given number of steps, where $\alpha$ is a Hebbian learning rate and $V \in \mathbb{R}^{NK \times K}$ is the matrix of postsynaptic responses $v_i$. We design the number of steps for which to run the update rule 1 as a hyperparameter of our method. Given a loss function $\mathcal{L}(\cdot, \cdot)$ and few-shot labels $y_i$, we choose the postsynaptic response

$$v_i = \nabla_{\hat{y}} \mathcal{L}(y_i, \hat{y})|_{\hat{y} = W z_i}, \tag{2}$$

casting the update rule 1 effectively as a gradient descent step. However, since we use only a few steps of rather strong updates we prefer to view this as a Hebbian learning rule. We initialize the weight matrix $W$ with zeros. In principle, any other initialization is possible but due to strong updates the initialization scheme is of minor importance.

**Combining several Hebbian Learners.** The closer a layer is to the network output the more specific are its features. Conversely, the closer a layer is to the input of the network, the more general are the features. In cross-domain few-shot learning, it is not a priori clear how specific or general the

features should be because this depends on how close the target domain is to the training domain. Therefore, we design our few-shot learning algorithm such that it can flexibly choose the specificity of the features depending on the current episode. We achieve this by representation fusion, where the Hebbian learning rule is applied to several layers at different levels of the backbone network in parallel. This yields a separate prediction for each level of abstraction. The final classification result is then obtained from the sum of logits arising from the respective Hebbian learners. A schematic view of CHEF is shown in Alg. 1.

---

**Algorithm 1** CHEF algorithm. The data matrix $X$ consists of input vectors $x_i$ and the label matrix $Y$ consists of the corresponding label vectors $y_i$. The function BB activates the backbone network up to a certain layer specified by an index $l$. $L$ is the set of indices specifying the layers used in the ensemble, $\mathcal{L}$ is the loss function of the few-shot learning task at hand. The function HEBBRULE executes $M$ steps of the Hebbian learning rule and yields a weight matrix $W$ that maps the feature vectors in $Z$ to vectors of length $K$, which are then used for $K$-fold classification. $\alpha$ is the Hebbian learning rate.

---

**Require:** $\alpha, M, \text{SOFTMAX}(\cdot), \mathcal{L}(\cdot, \cdot), \text{BB}(\cdot, \cdot)$
  **function** HEBBRULE$(X, Y, l)$
    $W \leftarrow 0, Z \leftarrow \text{BB}(X, l)$
    **for** $m \in \{1, \ldots, M\}$ **do**
      $V \leftarrow \nabla_{(ZW^\top)} \mathcal{L}(Y, \text{SOFTMAX}(ZW^\top))$
      $W \leftarrow W - \alpha V^\top Z$
    **end for**
    **return** $W$
  **end function**
  **function** ENSEMBLE$(X, Y, L)$
    $U = \sum_{l \in L} \text{BB}(X, l) \text{HEBBRULE}(X, Y, l)^\top$
    **return** SOFTMAX$(U)$
  **end function**

---

## 4 EXPERIMENTS

We apply CHEF to four cross-domain few-shot challenges, where we obtain state-of-the-art results in all categories. The four cross-domain few-shot challenges are characterized by domain shifts of different size, which we measure using the Fréchet-Inception-Distance (FID). We conduct ablation studies showing the influence of the different layer representations on the results. Further, we test CHEF on two standardized image-based few-shot classification benchmark datasets established in the field, which are characterized by a prior domain shift: *mini*Imagenet (Vinyals et al., 2016) and *tiered*Imagenet (Ren et al., 2018). Finally, we illustrate the impact of our CHEF approach on two real-world applications in the field of drug discovery, which are characterized first by a small domain shift and second by a large domain shift.

### 4.1 CROSS-DOMAIN FEW-SHOT LEARNING

**Dataset and evaluation.** The cross-domain few-shot learning challenge (Guo et al., 2019) uses *mini*Imagenet as training domain and then evaluates the trained models on four different test domains with increasing distance to the training domain: 1) CropDisease (Mohanty et al., 2016) consisting of plant disease images, 2) EuroSAT (Helber et al., 2019), a collection of satellite images, 3) ISIC2018 (Tschandl et al., 2018; Codella et al., 2019) containing dermoscopic images of skin lesions, and 4) ChestX (Wang et al., 2017) containing a set of X-ray images. For evaluation, we measure the accuracy drawing 800 tasks (five test instances per class) from the cross-domain test set. Following prior work, we focus on 5-way/5-shot, 5-way/20-shot, and 5-way/50-shot tasks. We report the average accuracy and a 95 % confidence interval across all test images and tasks.

**Measuring the domain shifts via FID.** In Guo et al. (2019), the four datasets of the new domain are characterized by their distance to the original domain using three criteria: whether images contain perspective distortion, the semantic content of images, and color depth. In Table 1, we provide measurements of the domain shift of these four datasets with respect to the original *mini*Imagenet

| Dataset | Conceptual difference to original domain (*mini*Imagenet) | FID |
|---|---|---|
| CropDisease | None | 257.58 |
| EuroSAT | No perspective distortion | 151.64 |
| ISIC2018 | No perspective distortion, unnatural content | 294.05 |
| ChestX | No perspective distortion, unnatural content, different color depth | 312.52 |

Table 1: Conceptual difference and domain shift between *mini*Imagenet and the four cross-domain datasets CropDisease, EuroSAT, ISIC2018, and ChestX. The domain shift is measured using the FID.

dataset using the FID. The FID measurements confirm the characterization in Guo et al. (2019), except that the EuroSAT dataset is closer to the original domain than the CropDisease dataset. The difference in both FID measurements is mostly driven by the mean terms. This can be explained by the fact that the FID does not measure perspective distortion and satellite images might have a higher variety of shapes and colors than plant images.

**CHEF implementation.** We perform pre-training on the *mini*Imagenet dataset similar but not identical to that in Ye et al. (2020). We utilize a softmax output layer with as many units as classes are contained in the meta-training and the meta-validation sets combined. We make a validation split on the combination of these two sets for supervised learning, i.e. instead of separating whole classes into the validation set (vertical split) we move a randomly selected fraction of samples of each class into the validation set (horizontal split) as it is standard in supervised learning. We evaluate CHEF using the same ResNet-10 backbone architecture as in Guo et al. (2019). For better representation fusion, we place two fully connected layers after the last convolutional layer. We perform model selection during training using the cross-entropy loss function on the horizontal data split, and perform hyperparameter selection for CHEF on the vertical data split.

**Results and ablation study.** CHEF achieves state-of-the-art performance in all 12 categories. Results are provided in Table 2. To further study the influence and power of the representation fusion, we use a pre-trained PyTorch (Paszke et al., 2019) ResNet-18 network. 5-way 5-shot and 50-shot results are reported in Fig. 2 (5-way 20-shot results can be found in the appendix). Results are obtained by applying our Hebbian learning rule to the logits of the output layer and to the pre-activations of the blocks 4 through 8 individually and we also examine an ensemble of them. The results are considerably better than the above reported ResNet-10 results, which presumably arises from the fact that the power of representation fusion is larger since the ResNet-18 network is pretrained on the whole Imagenet dataset. This illustrates the power of CHEF considering better feature abstractions. Another interesting insight is that for the ChestX dataset, the dataset with the largest domain shift, the lower level features gain importance. In general, the farther the domain is away from the original domain the more important are features from lower layers, i.e. features that are less specific to the original domain. Since CHEF combines features of different specificity to the training domain, it is particularly powerful in cross-domain settings.

## 4.2 MINIIMAGENET AND TIEREDIMAGENET

**Datasets and evaluation.** The *mini*Imagenet dataset (Vinyals et al., 2016) consists of 100 randomly chosen classes from the ILSVRC-2012 dataset (Russakovsky et al., 2015). We use the commonly-used class split proposed in Ravi & Larochelle (2017). The *tiered*Imagenet dataset (Ren et al., 2018) is a subset of ILSVRC-2012 (Russakovsky et al., 2015), composed of 608 classes grouped in 34 high-level categories. For evaluation, we measure the accuracy drawing 800 tasks (five test instances per class) from the meta-test set. Following prior work, we focus on 5-way/1-shot and 5-way/5-shot tasks. We report the average accuracy and a 95 % confidence interval across all test images and tasks.

**CHEF implementation and results.** We perform pre-training of the respective backbone networks on the *mini*Imagenet and the *tiered*Imagenet dataset in the same way as described in Sec. 4.1. We evaluate CHEF using two different backbone architectures: a Conv-4 and a ResNet-12 network. We use the Conv-4 network described by Vinyals et al. (2016). Following Lee et al. (2019), we configure the ResNet-12 backbone as 4 residual blocks, which contain a max-pooling and a batch-norm layer and are regularized by DropBlock (Ghiasi et al., 2018). Again, model selection and hyper-parameter tuning is performed as described in Sec. 4.1. CHEF achieves state-of-the-art performance in 5

| | CropDiseases 5-way | | | EuroSAT 5-way | | |
|---|---|---|---|---|---|---|
| **Method** | **5-shot** | **20-shot** | **50-shot** | **5-shot** | **20-shot** | **50-shot** |
| MatchingNet[†] | $66.39 \pm 0.78$ | $76.38 \pm 0.67$ | $58.53 \pm 0.73$ | $64.45 \pm 0.63$ | $77.10 \pm 0.57$ | $54.44 \pm 0.67$ |
| MatchingNet+FWT[†] | $62.74 \pm 0.90$ | $74.90 \pm 0.71$ | $75.68 \pm 0.78$ | $56.04 \pm 0.65$ | $63.38 \pm 0.69$ | $62.75 \pm 0.76$ |
| MAML[†] | $78.05 \pm 0.68$ | $89.75 \pm 0.42$ | - | $71.70 \pm 0.72$ | $81.95 \pm 0.55$ | - |
| ProtoNet[†] | $79.72 \pm 0.67$ | $88.15 \pm 0.51$ | $90.81 \pm 0.43$ | $73.29 \pm 0.71$ | $82.27 \pm 0.57$ | $80.48 \pm 0.57$ |
| ProtoNet+FWT[†] | $72.72 \pm 0.70$ | $85.82 \pm 0.51$ | $87.17 \pm 0.50$ | $67.34 \pm 0.76$ | $75.74 \pm 0.70$ | $78.64 \pm 0.57$ |
| RelationNet[†] | $68.99 \pm 0.75$ | $80.45 \pm 0.64$ | $85.08 \pm 0.53$ | $61.31 \pm 0.72$ | $74.43 \pm 0.66$ | $74.91 \pm 0.58$ |
| RelationNet+FWT[†] | $64.91 \pm 0.79$ | $78.43 \pm 0.59$ | $81.14 \pm 0.56$ | $61.16 \pm 0.70$ | $69.40 \pm 0.64$ | $73.84 \pm 0.60$ |
| MetaOpt[†] | $68.41 \pm 0.73$ | $82.89 \pm 0.54$ | $91.76 \pm 0.38$ | $64.44 \pm 0.73$ | $79.19 \pm 0.62$ | $83.62 \pm 0.58$ |
| CHEF (Ours) | $\mathbf{86.87 \pm 0.27}$ | $\mathbf{94.78 \pm 0.12}$ | $\mathbf{96.77 \pm 0.08}$ | $\mathbf{74.15 \pm 0.27}$ | $\mathbf{83.31 \pm 0.14}$ | $\mathbf{86.55 \pm 0.15}$ |
| | ISIC 5-way | | | ChestX 5-way | | |
| **Method** | **5-shot** | **20-shot** | **50-shot** | **5-shot** | **20-shot** | **50-shot** |
| MatchingNet[†] | $36.74 \pm 0.53$ | $45.72 \pm 0.53$ | $54.58 \pm 0.65$ | $22.40 \pm 0.7$ | $23.61 \pm 0.86$ | $22.12 \pm 0.88$ |
| MatchingNet+FWT[†] | $30.40 \pm 0.48$ | $32.01 \pm 0.48$ | $33.17 \pm 0.43$ | $21.26 \pm 0.31$ | $23.23 \pm 0.37$ | $23.01 \pm 0.34$ |
| MAML[†] | $40.13 \pm 0.58$ | $52.36 \pm 0.57$ | - | $23.48 \pm 0.96$ | $27.53 \pm 0.43$ | - |
| ProtoNet[†] | $39.57 \pm 0.57$ | $49.50 \pm 0.55$ | $51.99 \pm 0.52$ | $24.05 \pm 1.01$ | $28.21 \pm 1.15$ | $29.32 \pm 1.12$ |
| ProtoNet+FWT[†] | $38.87 \pm 0.52$ | $43.78 \pm 0.47$ | $49.84 \pm 0.51$ | $23.77 \pm 0.42$ | $26.87 \pm 0.43$ | $30.12 \pm 0.46$ |
| RelationNet[†] | $39.41 \pm 0.58$ | $41.77 \pm 0.49$ | $49.32 \pm 0.51$ | $22.96 \pm 0.88$ | $26.63 \pm 0.92$ | $28.45 \pm 1.20$ |
| RelationNet+FWT[†] | $35.54 \pm 0.55$ | $43.31 \pm 0.51$ | $46.38 \pm 0.53$ | $22.74 \pm 0.40$ | $26.75 \pm 0.41$ | $27.56 \pm 0.40$ |
| MetaOpt[†] | $36.28 \pm 0.50$ | $49.42 \pm 0.60$ | $54.80 \pm 0.54$ | $22.53 \pm 0.91$ | $25.53 \pm 1.02$ | $29.35 \pm 0.99$ |
| CHEF (Ours) | $\mathbf{41.26 \pm 0.34}$ | $\mathbf{54.30 \pm 0.34}$ | $\mathbf{60.86 \pm 0.18}$ | $\mathbf{24.72 \pm 0.14}$ | $\mathbf{29.71 \pm 0.27}$ | $\mathbf{31.25 \pm 0.20}$ |

[†] Results reported in Guo et al. (2019)

Table 2: Comparative results of few-shot learning methods on four proposed cross-domain few-shot challenges CropDiseases, EuroSAT, ISIC, and ChestX. The average 5-way few-shot classification accuracies (%, top-1) along with 95% confidence intervals are reported on the test split of each dataset.

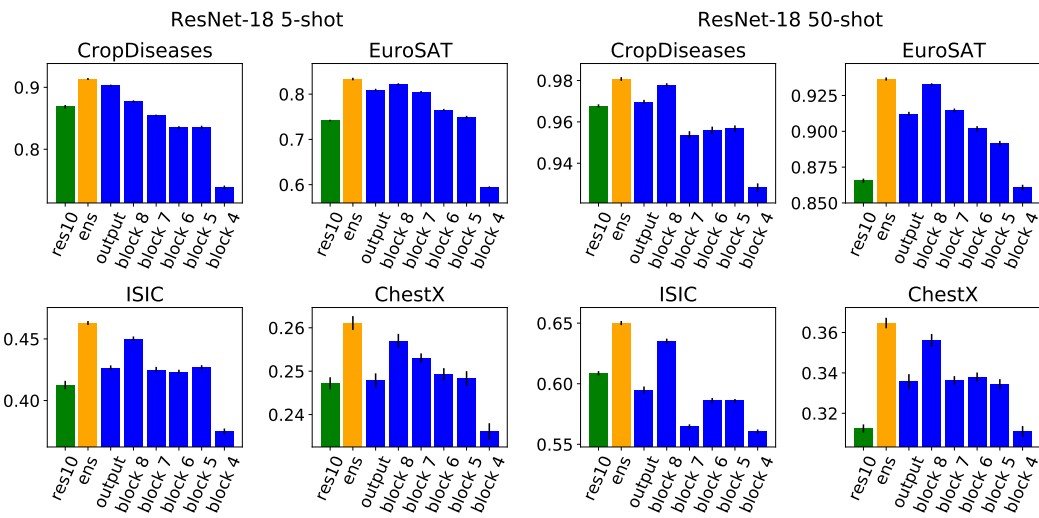

Figure 2: 5-shot and 50-shot top-1 accuracies (along with 95% confidence intervals) of different residual blocks and the output layer of an Imagenet-pretrained ResNet-18 and the ensemble result (orange, "ens") on the four different datasets of the cross-domain few-shot learning benchmark. For comparison, also the ResNet-10 ensemble results (green) are included.

out of 8 categories. Results are provided in Table 3. An ablation study of the *mini*Imagenet and *tiered*Imagenet results can be found in the appendix.

## 4.3 EXAMPLE APPLICATION: DRUG DISCOVERY

In drug discovery, it is essential to know properties of drug candidates, such as biological activities or toxicity (Sturm et al., 2020; Mayr et al., 2018). Since the measurements of these properties require time- and cost-intensive laboratory experiments, machine learning models are used to substitute such

| Method | Backbone | *mini*Imagenet 5-way | | *tiered*Imagenet 5-way | |
|---|---|---|---|---|---|
| | | 1-shot | 5-shot | 1-shot | 5-shot |
| MatchingNet (Vinyals et al., 2016) | Conv-4 | $43.56 \pm 0.84$ | $55.31 \pm 0.73$ | - | - |
| Meta-LSTM (Ravi & Larochelle, 2017) | Conv-4 | $43.44 \pm 0.77$ | $60.60 \pm 0.71$ | | |
| MAML (Finn et al., 2017) | Conv-4 | $48.70 \pm 1.84$ | $63.11 \pm 0.92$ | $51.67 \pm 1.81^{\dagger}$ | $70.30 \pm 1.75^{\dagger}$ |
| ProtoNets (Snell et al., 2017a) | Conv-4 | $49.42 \pm 0.78$ | $68.20 \pm 0.66$ | $48.58 \pm 0.87^{\dagger}$ | $69.57 \pm 0.75^{\dagger}$ |
| Reptile (Nichol et al., 2018) | Conv-4 | $47.07 \pm 0.26$ | $62.74 \pm 0.37$ | $48.97 \pm 0.21^{\dagger}$ | $66.47 \pm 0.21^{\dagger}$ |
| RelationNet (Sung et al., 2018) | Conv-4 | $50.44 \pm 0.82$ | $65.32 \pm 0.70$ | $54.48 \pm 0.93^{\dagger}$ | $71.32 \pm 0.78^{\dagger}$ |
| IMP (Allen et al., 2019) | Conv-4 | $49.60 \pm 0.80$ | $68.10 \pm 0.80$ | - | - |
| FEAT (Ye et al., 2020) | Conv-4 | $55.15 \pm 0.20$ | $71.61 \pm 0.16$ | - | - |
| Dynamic FS (Gidaris & Komodakis, 2018) | Conv-4 | $56.20 \pm 0.86$ | $72.81 \pm 0.62$ | - | - |
| CHEF (Ours) | Conv-4 | $\mathbf{57.60 \pm 0.29}$ | $\mathbf{73.26 \pm 0.23}$ | $\mathbf{61.10 \pm 0.21}$ | $\mathbf{75.83 \pm 0.25}$ |
| SNAIL (Mishra et al., 2018) | ResNet-12 | $55.71 \pm 0.99$ | $68.88 \pm 0.92$ | - | - |
| TADAM (Oreshkin et al., 2018) | ResNet-12 | $58.50 \pm 0.30$ | $76.70 \pm 0.30$ | - | - |
| MTL (Sun et al., 2019) | ResNet-12 | $61.20 \pm 1.80$ | $75.50 \pm 0.80$ | - | - |
| VariationalFSL (Zhang et al., 2019) | ResNet-12 | $61.23 \pm 0.26$ | $77.69 \pm 0.17$ | - | - |
| TapNet (Yoon et al., 2019) | ResNet-12 | $61.65 \pm 0.15$ | $76.36 \pm 0.10$ | $63.08 \pm 0.15$ | $80.26 \pm 0.12$ |
| MetaOptNet (Lee et al., 2019) | ResNet-12 | $62.64 \pm 0.61$ | $78.63 \pm 0.46$ | $65.81 \pm 0.74$ | $81.75 \pm 0.53$ |
| CTM (Li et al., 2019) | ResNet-12 | $62.05 \pm 0.55$ | $78.63 \pm 0.06$ | $64.78 \pm 0.11$ | $81.05 \pm 0.52$ |
| CAN (Hou et al., 2019) | ResNet-12 | $63.85 \pm 0.48$ | $79.44 \pm 0.34$ | $69.89 \pm 0.51$ | $84.23 \pm 0.37$ |
| FEAT (Ye et al., 2020) | ResNet-12 | $\mathbf{66.78 \pm 0.20}$ | $\mathbf{82.05 \pm 0.14}$ | $\mathbf{70.80 \pm 0.23}$ | $84.79 \pm 0.16$ |
| Dynamic FS (Gidaris & Komodakis, 2018) | ResNet-12 | $55.45 \pm 0.89$ | $70.13 \pm 0.68$ | - | - |
| CHEF (Ours) | ResNet-12 | $64.11 \pm 0.32$ | $79.99 \pm 0.21$ | $70.70 \pm 0.35$ | $\mathbf{85.97 \pm 0.09}$ |

[†] Results reported in (Liu et al., 2019)

Table 3: Comparative results of few-shot learning methods on the two benchmark datasets *mini*Imagenet and *tiered*Imagenet. The average 5-way few-shot classification accuracies (%, top-1) along with 95% confidence intervals are reported on the test split of each dataset.

measurements (Hochreiter et al., 2018). However, due to the high experimental effort often only few high-quality measurements are available for training. Thus, few-shot learning is highly relevant for computational drug discovery.

**Problem setting.** We consider a 50-shot cross-domain few-shot learning setting in the field of toxicity prediction, utilizing the *Tox21 Data Challenge dataset* (Tox21) with twelve different toxic effects (Huang & Xia, 2017; Mayr et al., 2016). Around 50 available measurements is a typical scenario when introducing a new high-quality assay in drug design. So far, the standard approach to deal with such few data points was to use machine learning methods like Support Vector Machines (SVMs; Cortes & Vapnik, 1995) or Random Forests (RFs; Breiman, 2001). However, these methods do not exploit the rich data available, like the *ChEMBL20 drug discovery benchmark* (ChEMBL20) (Mayr et al., 2018; Gaulton et al., 2017). Viewing the Tox21 data as a domain shift of the ChEMBL20 allows the application of cross-domain few-shot learning methods. In this setting, a domain shift can be observed both in the input data and in the target data. The molecules (input domain) are strongly shifted towards a specialized chemical space, with a Jaccard index of 0.01 between the two datasets, and the biological effects (output domain) are shifted towards toxicity without any overlap in this domain. A further shift is in the distribution of the target labels, which are now much more imbalanced in comparison to ChEMBL20. In order to mirror this distribution shift correctly, the number of toxic vs. non-toxic molecules in the training sets for each of the twelve few-shot tasks (twelve different toxic effects) are sub-sampled accordingly. For example, the 50-shot few-shot scenario (50-50 toxic/non-toxic) is adjusted to a 10-90 scenario. For the twelve few-shot learning tasks, training samples are drawn from the training set and test samples are drawn from the test set of the Tox21 data, respectively. We sample individually for each of the twelve tasks.

**CHEF implementation for molecules.** We first train a fully-connected deep neural network (FCN) for the prediction of bioactivities from the ChEMBL20 database (original domain). The network is trained in a massive multi-task setting, where 1,830 tasks are predicted at once, such that the network is forced to learn proficient representations that can be shared for multiple tasks (Ma et al., 2015; Unterthiner et al., 2014). The total number of 892,480 features of the ChEMBL20 database was reduced by a sparseness criterion on the molecules to 1,866 features. The neurons in the input layer of the FCN represent one of 1,866 ECFP6 (Rogers & Hahn, 2010) features, which are used as a feature representation for describing the raw structure of the molecules. Each neuron of the output layer represents one of the 1,830 prediction tasks. We use the pre-trained network and apply CHEF by representation fusion of the three bottleneck layers of the network for predicting the twelve different toxic effects of the new domain of the Tox21 Data Challenge.

**Experimental evaluation.** We evaluate the performance of CHEF on the twelve tasks of the Tox21 Data Challenge and compare it to conventional methods, like SVMs and RFs, that are used in drug design when few data is available. We use SVMs with a MinMax kernel since it previously yielded the best results (Mayr et al., 2018). For CHEF, only the 1,866 ECFP input features of ChEMBL20 pre-training network database are used where features with only few occurrences in the training set are discarded since they do not give enough learning signal for neural network training. For SVMs and RFs, all encountered ECFP features are used. ROC-AUC values are computed across the twelve tasks of the Tox21 Data Challenge and across 100 differently sampled training and test sets. CHEF achieves significantly better ROC-AUC values than SVMs and RFs. Table 4 shows

| Method | ROC-AUC |
|--------|---------|
| CHEF | **0.76** $\pm$ 0.02 |
| SVM | 0.66 $\pm$ 0.03 |
| RF | 0.64$\pm$ 0.03 |

Table 4: ROC-AUC performance for few-shot drug discovery. CHEF is compared to conventional methods (SVM, RF) for the prediction of toxic effects. Mean and standard deviation are computed across twelve different effects and across 100 differently sampled training and test sets.

the results ($p$-value $< 10^{-17}$ for both SVM and RF when using a paired Wilcoxon test). Results for the twelve individual tasks and a more detailed description is given in the appendix. CHEF significantly outperforms traditional methods in drug design, which demonstrates the great potential of cross-domain few-shot learning in this field.

## 5 CONCLUSION

We have introduced CHEF as new cross-domain few-shot learning method. CHEF builds on the concept of representation fusion, which unifies information from different levels of abstraction. Representation fusion allows one to successfully tackle various few-shot learning problems with large domain shifts across a wide range of different tasks. CHEF obtains new state-of-the-art results in all categories of the broader study of cross-domain few-shot learning benchmarks. Finally, we have tested the performance of CHEF in a real-world cross-domain application in drug discovery, i.e. toxicity prediction when a domain shift appears. CHEF significantly outperforms all traditional approaches demonstrating great potential for applications in computational drug discovery.

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

## A  APPENDIX

### A.1  EXPERIMENTAL SETUP

In the following, we give further details on our experimental setups.

#### A.1.1  CROSS-DOMAIN FEW-SHOT LEARNING

We utilize a ResNet-10 backbone architecture as proposed in Guo et al. (2019). The residual blocks have 64, 128, 256, 512, 4000, and 1000 units, where the latter two are fully connected ReLU layers. We use a learning rate of 0.1, momentum term of 0.9, L2 weight decay term of $10^{-4}$, batch size of 256, dropout rate of 0.5 during pre-training. These values were tuned on the horizontal validation set of *mini*Imagenet. For few-shot learning, we choose a Hebbian learning rate of $\alpha = 0.01$ and run the Hebb rule for $I = 400$ steps. These values were tuned on the vertical validation set of *mini*Imagenet.

Figures 2 and 3 show the performance of the pre-trained PyTorch ResNet-18 network, where the pre-training is on the entire Imagenet dataset. Additionally, the individual performances of the ResNet-18 layers are depicted. The *mini*Imagenet pre-trained ResNet-10 is shown for comparison. The plots show the general tendency that the ensemble performance on domains which are farther away from the training domain relies more heavily on features in lower layers, i.e. features with less specificity to the original domain.

| Method | CropDiseases 5-way | | | EuroSAT 5-way | | |
|---|---|---|---|---|---|---|
| | 5-shot | 20-shot | 50-shot | 5-shot | 20-shot | 50-shot |
| CHEF (ResNet-10) | $86.87 \pm 0.27$ | $94.78 \pm 0.12$ | $96.77 \pm 0.08$ | $74.15 \pm 0.27$ | $83.31 \pm 0.14$ | $86.55 \pm 0.15$ |
| CHEF (ResNet-18) | $91.34 \pm 0.16$ | $96.99 \pm 0.07$ | $98.07 \pm 0.09$ | $83.44 \pm 0.28$ | $91.62 \pm 0.13$ | $93.65 \pm 0.11$ |
| Method | ISIC 5-way | | | ChestX 5-way | | |
| | 5-shot | 20-shot | 50-shot | 5-shot | 20-shot | 50-shot |
| CHEF (ResNet-10) | $41.26 \pm 0.34$ | $54.30 \pm 0.34$ | $60.86 \pm 0.18$ | $24.72 \pm 0.14$ | $29.71 \pm 0.27$ | $31.25 \pm 0.20$ |
| CHEF (ResNet-18) | $46.29 \pm 0.16$ | $58.85 \pm 0.26$ | $65.01 \pm 0.17$ | $26.11 \pm 0.16$ | $31.83 \pm 0.35$ | $36.47 \pm 0.26$ |

Table 5: Results of our few-shot learning method CHEF on four proposed cross-domain few-shot challenges CropDiseases, EuroSAT, ISIC, and ChestX. We compare the ResNet-10 architecture pre-trained on *mini*Imagenet to the ResNet-18 architecture pre-trained on Imagenet. The average 5-way few-shot classification accuracies (%, top-1) along with 95% confidence intervals are reported on the test split of each dataset.

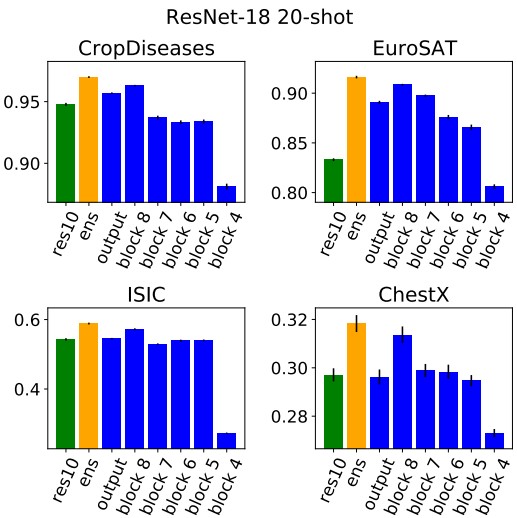

Figure 3: 20-shot top-1 accuracies (along with 95% confidence intervals) of different residual blocks and the output layer of an Imagenet-pretrained ResNet-18 and the ensemble result (orange, "ens") on the four different datasets of the cross-domain few-shot learning benchmark. For comparison, also the ResNet-10 ensemble results (green) are included.

### A.1.2    MINIIMAGENET AND TIEREDIMAGENET

**Backbone pre-training.**    For the *mini*Imagenet and *tiered*Imagenet experiments, we utilize Conv-4 and ResNet-12 architectures as backbone networks. The Conv-4 network is described in detail by Vinyals et al. (2016). It is a stack of 4 modules, each of which consists of a $3 \times 3$ convolutional layer with 64 units, a batch normalization layer (Ioffe & Szegedy, 2015), a ReLU activation and $2 \times 2$ max-pooling layer. On top, we place two fully connected ReLU layers with 400 and 100 units, respectively. The ResNet-12 is described in Lee et al. (2019). We configure the backbone as 4 residual blocks with 64, 160, 320, 640, 4000, and 1000 units, respectively, where the latter two are ReLU-activated fully connected layers. The residual blocks contain a max-pooling and a batch-norm layer and are regularized by DropBlock (Ghiasi et al., 2018) with block sizes of $1 \times 1$ for the first two blocks and $5 \times 5$ for the latter two blocks.

We pre-train these backbone models for 500 epochs with three different learning rates 0.1, 0.01, and 0.001. For this we use the PyTorch SGD module for stochastic gradient descent with a momentum term of 0.9, an L2 weight decay factor of $10^{-4}$, a mini-batchsize of 256, and a dropout probability of 0.1. This pre-training is performed on the horizontal training set of the *mini*Imagenet and the *tiered*Imagenet dataset, resulting in 3 trained models per dataset. We apply early stopping by selecting the model with the lowest loss on the horizontal validation set, while evaluating the model performance after each epoch.

| parameter | values |
|---|---|
| learning rate of pre-trained model | $\{0.1, 0.01, 0.001\}$ |
| dropout probability | $0.5$ |
| Hebbian learning rate | $\{0.1, 0.01, 0.001\}$ |
| number of Hebb rule steps | $\{1, 2, 5, 7, 10, 25, 50, 75, 100, 250, 500, 750\}$ |

Table 6: Hyper-parameter search space for 1-shot and 5-shot learning on *mini*Imagenet and *tiered*Imagenet using Conv-4 and ResNet-12 backbone models. Best hyper-parameters were evaluated using a grid-search and the loss on the vertical validation set of *mini*Imagenet or *tiered*Imagenet.

**Few-shot learning.** For few-shot learning, we perform a grid search to determine the best hyper-parameter setting for each of the datasets and each of the 1-shot and 5-shot settings, using the loss on the vertical validation set. We treat the 3 backbone models that were pre-trained with different learning rates, as described in the previous paragraph, as hyper-parameters. The hyper-parameters used for this grid-search are listed in table 6.

After determining the best hyper-parameter setting following this procedure, we perform 1-shot and 5-shot learning on the vertical test sets of *mini*Imagenet and *tiered*Imagenet using 10 different random seeds, respectively. The results are listed in table 3.

**Ensemble learning and performance of individual layers.** To evaluate the performance of the Hebbian learning using only individual layers versus using the ensemble of layers, we additionally perform the few-shot learning on the vertical test sets using only individual layers as input to the Hebbian learning. As shown in figures 4 and 5, the performance using only the individual layers varies strongly throughout 1-shot and 5-shot learning and the *mini*Imagenet and *tiered*Imagenet dataset. This indicates that the usefulness of the representations provided by the individual layers strongly depends on the data and task setting. In contrast to this, the ensemble of layers reliably achieves either best or second best performance throughout all settings.

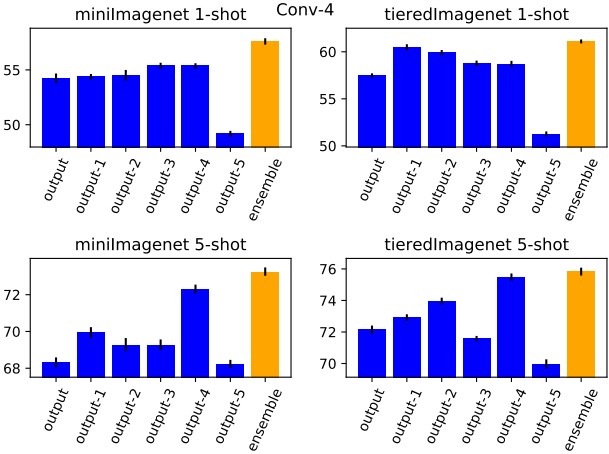

Figure 4: Ablation study of the Conv-4 architecture on the *mini*Imagenet and *tiered*Imagenet datasets for 1-shot and 5-shot. The plots show the individual performances of Hebbian learners acting on single layers and their ensemble performance along with 95% confidence intervals. The labels on the $x$-axis indicate how far the respective layer is from the output layer.

### A.1.3 EXAMPLE APPLICATION: DRUG DISCOVERY

**Details on pre-training on the ChEMBL20 database** For training a fully-connected deep neural network (FCN) on the ChEMBL20 database, the number of 892,480 features is reduced by a sparseness criterion on the molecules to 1,866 features. The FCN is trained on 1.1 million molecules for 1,000 epochs minimizing binary cross-entropy and masking out missing values by using an objective, as described in (Mayr et al., 2018).

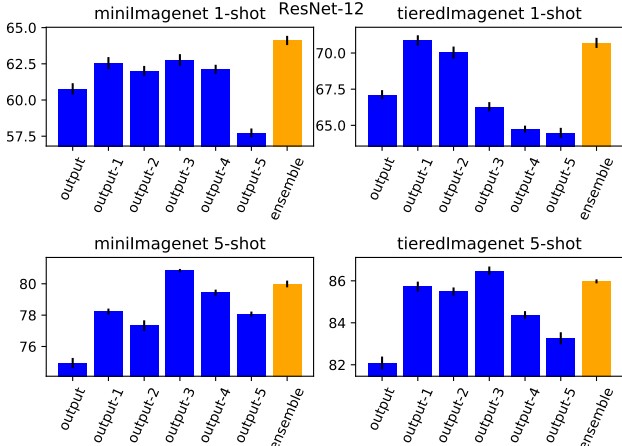

Figure 5: Ablation study of the ResNet-12 architecture on the *mini*Imagenet and *tiered*Imagenet datasets for 1-shot and 5-shot. The plots show the individual performances of Hebbian learners acting on single layers and their ensemble performance along with 95% confidence intervals. The labels on the $x$-axis indicate how far the respective layer is from the output layer.

**Details on the compared methods**    We use ECFP6 features for a raw molecule representation. Note that the number of possible distinct ECFP6 features is not predefined, since a new molecule may be structurally different from all previously seen ones, and it might therefore consist of new unseen ECFP6 features. For SVMs, a MinMax kernel (Mayr et al., 2016) is used, which operates directly on counts of ECFP6 features and used LIBSVM (Chang & Lin, 2011) as provided by scikit-learn (Pedregosa et al., 2011). For RFs, the implementation of scikit-learn with 1000 trees and kept default values for the other hyperparameters is used.

**Detailed results on the Tox21 dataset in a few-shot setup**    Table 7 lists detailed results and $p$-values of all twelve few-shot tasks of the Tox21 Data Challenge. For calculating $p$-values, a paired Wilcoxon test is used.

| Dataset | CHEF | SVM | RF | p-value SVM | p-value RF |
|---|---|---|---|---|---|
| NR.AhR | **0.86 ± 0.07** | 0.79 ± 0.07 | 0.75 ± 0.07 | 2.90e-12 | 1.19e-17 |
| NR.AR | **0.79 ± 0.09** | 0.60 ± 0.11 | 0.61 ± 0.11 | 1.20e-17 | 5.25e-18 |
| NR.AR.LBD | **0.84 ± 0.05** | 0.47 ± 0.11 | 0.52 ± 0.10 | 1.94e-18 | 1.95e-18 |
| NR.Aromatase | **0.74 ± 0.08** | 0.68 ± 0.09 | 0.64 ± 0.09 | 3.77e-09 | 1.12e-13 |
| NR.ER | **0.73 ± 0.08** | 0.70 ± 0.08 | 0.65 ± 0.09 | 1.39e-03 | 4.25e-11 |
| NR.ER.LBD | **0.71 ± 0.08** | 0.68 ± 0.09 | 0.65 ± 0.10 | 1.96e-03 | 2.40e-06 |
| NR.PPAR.gamma | **0.66 ± 0.07** | 0.61 ± 0.10 | 0.60 ± 0.11 | 8.04e-06 | 3.05e-06 |
| SR.ARE | **0.76 ± 0.08** | 0.66 ± 0.08 | 0.61 ± 0.09 | 2.43e-14 | 2.51e-17 |
| SR.ATAD5 | **0.68 ± 0.07** | 0.62 ± 0.10 | 0.61 ± 0.10 | 2.23e-07 | 7.65e-10 |
| SR.HSE | **0.74 ± 0.06** | 0.62 ± 0.10 | 0.60 ± 0.10 | 3.42e-16 | 1.40e-16 |
| SR.MMP | **0.89 ± 0.05** | 0.81 ± 0.08 | 0.79 ± 0.09 | 7.36e-15 | 4.41e-16 |
| SR.p53 | **0.77 ± 0.08** | 0.67 ± 0.10 | 0.63 ± 0.10 | 3.50e-13 | 8.86e-17 |

Table 7: ROC-AUC performances for the twelve individual few-shot tasks (rows) of the Tox21 Data Challenge. CHEF is compared to conventional methods (SVM, RF). Averages and standard deviations are computed across 100 differently sampled training and test sets. The last two columns show the results of paired Wilcoxon tests with the null hypotheses given that SVM and RF perform better, respectively.

