# OpenReview forum: "Cross-Domain Few-Shot Learning by Representation Fusion"
_ICLR.cc/2021/Conference — Reject_

### Official Review · AnonReviewer4 · 2020-10-26
**CROSS-DOMAIN FEW-SHOT LEARNING BY REPRESENTATION FUSION**

**Rating:** 4
**Confidence:** 5

**Review:**

Summary:  This paper proposes a domain-shift problem using fewer training examples. It suggests representation fusion as the concept of unifying and merging information from different layers of abstraction.
Cross-domain Hebbian Ensemble Few-shot learning (CHEF) is introduced for extracting features using representation fusion.  More importantly, CHEF does not need to backpropagate information through the backbone network.   CHEF  is applied to various cross-domain few-shot tasks and cross-domain real-world applications from drug discovery.

Strong Points: 1-  The paper is well organized and easy to understand.
2-  The model is evaluated in four benchmark datasets CropDisease, EuroSAT,  ISIC2018, and ChestX.  It also conducted experiments on two large scale datasets (prepared from ImageNet dataset), miniImagenet and tieredImagenet.  The proposed model shows consistent and promising performance in all datasets.

3- It introduced Hebbian learners for feature fusion that does not require backpropagation of error signals through the entire backbone network.  Only the parameters of the Hebbian learners need adjustment. Therefore it is speedy and versatile.

Weaknesses: 1- The crucial contribution of this work is Hebbian Learner. Therefore it should devote more space for Hebbian Learner. The explanation about Hebbian Learner provided in this paper is not sufficient understanding to propose an approach clearly. I recommend the authors that include more description of Hebbian Learner.  I understand the space limit, but the model's crucial and essential contribution should be included in the main paper.
2- This paper claims that using "Hebbian Learner makes CHEF extremely fast," but I did not find any supporting experiments in the main paper to prove this statement. It should include results on time comparison.
3- This paper performed the experiments on a few-shot learning setup for that chosen 5, 20, and 50 examples per class to train the model and observed continuous improvement on model performance. To show the limit of model performance, the experiments should also be performed using all available examples in the datasets.
4-  authors have selected ResNet-12 as the backbone model; any specific reason for this? I wonder to see the model performance for more deep networks like ResNet-101 as the backbone model.
5- It would be better to show the individual contribution of the Hebbian Learner. Therefore result should also be included in the ablation analysis section without using Hebbian Learner in the same setting.
6-   On page#4, the authors have mentioned that "We combine the NK feature vectors into a matrix Z ∈ R^ NK×D and initialize a weight matrix W ∈ R ^K×D."  But it is not mentioned about the initialization technique. Random initialization, Xavier initialization, etc. ?
7- Some essential baseline approaches are missing for comparison, such as  "Few-Shot Adversarial Domain Adaptation" by Saeid Motiian et al. NIPS 2017.


Overall: The paper needs to include many things for better clarity. I feel it provides some insufficient information or does not clearly explain the proposed model's crucial contribution. It needs to include experimental results for different settings, as I mentioned in the weaknesses section, to prove the model's efficacy.

---

> ### Author Response · Authors · 2020-11-17
> **Response to Reviewer 4**
>
> Thank you for a very thorough and helpful review. Please, let us expand a bit on the weaknesses you pointed out.
>
> 1. This is an excellent point. We extended the methods section by a discussion on how our Hebbian learning rule relates to gradient descent. However, we want to emphasize that we also consider the concept of representation fusion as our main contribution.
> 2. Most few-shot learning methods adjust the weights of the backbone network, i.e. the computational costs can be divided into three parts: (i) backbone inference, (ii) learner forward overhead (iii) backprop through (ii) and (i). Our method only needs to perform (i) and (ii) but not (iii). We could back this argument up with empirical evidence. However, we consider the fastness of our method an appreciated side effect rather than a main selling point (which is bridging large domain gaps).
> 3. This is a very good and interesting idea. To be comparable, we choose the same setting as in prior work [1]. Moreover, in this paper we focus on few-shot learning and therefore leave this point open for future work.
> 4. Yes, the reason for this is that the Res-12 architecture emerged as standard in the few-shot literature for reasons of comparability.
> 5. Yes, to work out the contributions of Hebbian learning on individual layers we conducted an ablation study which is shown in Fig. 2.
> 6. We initialize the Hebbian weights with zeros as described directly below Eq. (2).
> 7. Thank you for this very interesting reference. We will try our best to re-implement this method and run it on the relevant data sets as fast as we can.
>
> [1] Y. Guo, N. Codella, L. Karlinsky, J.R. Smith, T. Rosing, and R. Feris. A new benchmark for evaluation of cross-domain few-shot learning. arXiv preprint arXiv:1912.07200, 2019.

---

> ### Comment · AnonReviewer4 · 2020-11-24
> **Post Rebuttal**
>
> The authors have partially addressed my concerns; still, I am not convinced with a clear contribution of individual components and short experiments to support the author's claims.  Reviewer-5 also raised the same issue: missing information about Hebbian learning, FID, which makes it challenging to understand this paper. In my view, this paper needs significant work to compare with robust baseline approaches and present a solid ablation analysis to show the individual contribution of each proposed component. Therefore, I recommend rejection in its present form.

---

### Official Review · AnonReviewer2 · 2020-10-28
**Details on Hebbian Ensemble Learning**

**Rating:** 5
**Confidence:** 3

**Review:**

This paper proposes cross-domain Hebbian ensemble few-shot learning or CHEF which achieves representation fusion by an ensemble of Hebbian learners acting on different layers of a deep neural network that was trained on the original domain and aims at learning from few examples, often by using already acquired knowledge. The experiments show results on miniImageNet and tieredImageNet, where the domain shift is small and also on the cross domain few-shot benchmarks with larger domain shifts. The paper also shows auxiliary experiments on drug discovery. I have the following comments on the paper:

major comments
1. How do you compare the Hebbian ensemble learning strategy with other ensemble learning strategies, such as random forest and boosting? Is it possible to do a comparison between those?

2. The equation (1) basically depicts the Hebbian learning rule where V is the matrix of postsynaptic responses v_i which are effectively the gradient of the loss in equation (2). I wonder how this rule (equation 1) differs in various layers? I also wonder what does combining several Hebbian learners mean in the case of few-shot learning?

3. Is it possible to plot the datasets with T-SNE or PCA and present it beside Table 1 for a clear understanding of the dataset characteristics? I understand those are approximate methods, but it is interesting to see coherence of the plot with FID.

Based on my current understanding and the above comments, I currently recommend the paper as "marginally below acceptance threshold". I would like to hear clarification on Hebbian ensemble learning.

minor comments
1. A reference to Hebbian ensemble learning will be useful.
2. A reference to Fréchet-Inception-Distance (FID) will also be useful.
3. In ICLR 2020, there were few works that proposed to learn mutual information from diverse domains. I think it is worth to provide to have a discussion on them.
(i) M. Federici et al., Learning Robust Representations via Multi-View Information Bottleneck, ICLR, 2020.
(ii) M. Tschannen et al., On Mutual Information Maximization for Representation Learning, ICLR, 2020.
4. In figure 2, the labels along the x axis are confusingly aligned. I think it is better to make them exactly perpendicular to the x axis.

---

> ### Author Response · Authors · 2020-11-17
> **Response to Reviewer 2**
>
> Thank you for your percipient review that allows us to improve our paper. Ad major comments:
>
> 1. Yes you are right, it is perfectly possible to compare the Hebbian ensemble learners to other ensemble learning techniques. The idea was to treat each of the layers as an output layer of the network (representation fusion). Therefore, a learning rule that adjusts a simple mapping to the label space was a very natural choice to us. In a very early stage we tried out several weak learners like k-nearest neighbors and clustering methods. However, the biggest performance gain was due to representation fusion.
> 2. This depends on the size of the domain gap. For small gaps, the highest layers will dominate the ensemble prediction because high-level concepts are still applicable in the target domain and produce strong postsynaptic responses. For large gaps, these signals become less specific and the ensemble prediction resorts to relying on more low-level concepts.
> 3. Yes, doing t-SNE or PCA on image data sets is technically possible. Specific image-domain distance measures like FID have proven to be much more aligned with human intuition of similarity among images as it relies on feature embeddings through an Inception v3 network.
>
> Ad minor comments:
>
> 1. In the context of few-shot learning, ensemble methods have been discussed in [1], Hebbian learning was explored in [2].
> 2. The FID was proposed in [3]. We added one more descriptive sentence.
> 3. Thank you very much for the helpful references. We incorporated them into our related-work section.
> 4. In case of acceptance we will change that in the camera-ready version.
>
> [1] N. Dvornik, C. Schmid, and J. Mairal. Diversity with cooperation: Ensemble methods for few-shot classification. InProc. IEEE Int. Conf. Comput. Vis. (ICCV), pp. 3723–3731, 2019.
>
> [2] T. Munkhdalai and A. Trischler. Metalearning  with hebbian fast weights. arXiv  preprint arXiv:1807.05076, 2018.
>
> [3] M. Heusel, H. Ramsauer, T. Unterthiner, B. Nessler, and S. Hochreiter. Gans trained by a two time-scale update rule converge to a local nash equilibrium. In Advances in neural information processing systems, pp. 6626–6637, 2017.

---

### Official Review · AnonReviewer1 · 2020-10-29
**Novelty is limited**

**Rating:** 4
**Confidence:** 4

**Review:**

This paper introduce a learning mechanism that combining few-shot domain adaptation with a Hebbian learning rule. Basically, the authors fused multiple layer feature representations in weak learner and ensemble the classification results. This approach is trivial. I would suggest the author can introduce the benefit or provide the reason a Hebbian learning can improve adaptation performance.

---

### Official Review · AnonReviewer3 · 2020-11-03
**Cross-Domain Few-Shot Learning by Representation Fusion**

**Rating:** 6
**Confidence:** 3

**Review:**

In this paper, the authors focus on cross-domain few-shot learning in the case of large source-target domain shifts. In particular, a new Cross-domain Hebbian Ensemble Few-shot (CHEF) learning method is proposed that performs representation fusion using an ensemble of Hebbian learners on different layers of a DNN trained on the source domain. The proposed CHEF method is validated on classification benchmark datasets with smaller domain shifts (miniImagenet and tieredImagenet) and larger domain shifts (drug discovery, ChEMBL20), and it can outperform related SOTA methods, especially with larger shifts.

+ The paper is clearly written and generally well organized, and all the key concepts are detailed, although I found some parts of Section 3 difficult to follow.  Even with Figure 1 and Algorithm 1, the paper was not easy to follow. The section on related work is very condensed, without much critical analysis. Therefore, it is not clear how their CHEF is motivated by challenges in literature.  The literature e review for methods on DA is not up to date, and not reflect SOTA methods on deep DA.

+ The code is made available as supplementary material, so the results in this paper should be reproducible by a reader.

+  The supplementary material provides additional information that should be useful to the reader (experimental setup and results).

+ The authors present many interesting results in Section 4, and they are for the most part convincing. They do present averages results over independent replications, using some cross-validation process. Using the FID to measure the domain shifts is excellent. However, I am not convinced about the results shown in Section 4.3. If I understand, Table 4 compares a deep NN (FCN).

+ CHEF with some conventional ML models (SVM and RF)?  This experiment needs some clarification. Their model could be also compared with SOA methods in terms of time and/or memory complexity.

---

> ### Author Response · Authors · 2020-11-17
> **Response to Reviewer 3**
>
> Thank you for a very insightful review that helps us to improve our paper. We updated the related-work section to include SOTA domain adaptation literature. Thank you also for the feedback on section 4.3. Having only 50 measurements is a typical setting in drug discovery. For such small datasets, SVM and RF are SOTA methods. Only by reinterpreting this setting as a cross-domain few-shot learning task we are able to incorporate large datasets for pretraining and successfully apply deep learning models here.

---

### Official Review · AnonReviewer5 · 2020-11-08
**Simple algorithm but missing many key details**

**Rating:** 4
**Confidence:** 4

**Review:**

Summary:

This paper primarily deals with cross-domain few-shot learning. Under this setting, there is a large shift in domain going from the meta-train dataset to the few-shot datasets. Inspired by previous work, the authors argue that high-level concepts might not be useful in this setting but low-level concepts like edges, textures and shapes can be utilized. They propose a Cross-domain Hebbian Ensemble Few-shot (CHEF) learner, that learns an ensemble of classifiers at multiple levels of a deep neural network, thus making use of both low and high level concepts. Experimental results show that CHEF does better, in most cases, than learning a separate classifier at a given level. They show results under the cross-domain and the standard few-shot setting.

Pros:
1. Utilizing low and high level concepts is a simple technique to boost few-shot learning performance.
2. CHEF does not require any updates to the weights of the model backbone. It learns additional weights for classification.

Cons:
1. The paper is missing details. The authors talk about Hebbian learning, FID, etc. but do not give details about it. The experimental set-up is missing information about how the models are trained and tested.
2. Is the proposed algorithm a Hebbian learner? The update in Equations 1 and 2 is a standard gradient descent update.
3. Using 2 fully-connected layers changes the model backbone from ResNet-x to a ResNet-(x+1). This should be clearly noted in Table 2 and 3.

Clarifications:
1. For the experiments, the softmax output layer has as many units as the number of classes in the meta-train and meta-validation sets combined. Does this mean that the pre-training is done on the sets combined? If so, this is not an apples-to-apples comparison. If not, the model does not know the difference between the classes in the meta-validation set. How does having these classes in the validation set help while pre-training?

Notes:
1. Mini-ImageNet and Tiered-ImageNet do involve general domain shifts. Even though p(x) does not change much going from meta-train dataset to the few-shot datasets, the samples seen in the two scenarios are disjoint.
2. The Appendix should be cleaned up.

---

> ### Author Response · Authors · 2020-11-17
> **Response to Reviewer 5**
>
> Thank you for a very elaborate review that helps us to improve our paper. We address the contra points as follows.
> 1. Yes, some explanations were a bit short. Therefore, we expanded a bit on Hebbian learning and FID in the new paper draft.
> 2. This is an excellent point. Indeed, gradient descent can be viewed as an instance of Hebbian learning. The new paper draft now sheds some light on this connection. The main reason why we consider our method a Hebbian learning rule is that we use very strong updates, which we do not iterate until convergence but for a fixed number of steps.
> 3. Yes, that is correct. Our method does use some additional weights. The main reason for adding two fully connected layers is to have more hierarchically structured features, not to add more capacity. Many of the competing methods also use additional weights in the same ballpark like e.g. [1] or [2].
>
> Clarifications:
>
> 1. Yes, our method combines meta-training and meta-validation sets. Subsequently, we split the combination of these two sets into a training and a validation set (standard supervised split, “horizontal” split). This is necessary because we pretrain the backbone architecture in a standard supervised fashion and not using a few-shot learning technique. Therefore, we also have to use a standard supervised validation split. The number of samples we use for training and validation, however, is equal to that in the few-shot learning setting.
>
> [1] H.J. Ye, H. Hu, D.C. Zhan, and F. Sha. Few-shot learning via embedding adaptation with set-to-set functions. InComputer Vision and Pattern Recognition (CVPR), 2020.
>
> [2] S. Gidaris and N. Komodakis. Dynamic few-shot visual learning without forgetting. In Proceedings of the IEEE Conference on Computer Vision and Pattern Recognition, pp. 4367–4375, 2018.

---

### Author Response · Authors · 2020-11-17
**Thank you for your feedback!**

We want to thank all reviewers for their constructive feedback. It helped us a lot to improve our paper. We hope to answer all questions and provide clarifications in individual responses to the respective reviewers. Further, we uploaded a rebuttal revision of our paper incorporating your sound suggestions.

---

### Comment · Area_Chair1 · 2020-11-20
**Please, check rebuttals and start discussion, if needed**

Dear Reviewers and Authors,
Thanks for starting the discussion.

Reviewers: please, check the rebuttals provided by the authors, verify if they replied properly and you are satisfied.
Possibly, give further feedback or make questions, only if needed and important for your final evaluation.
Please, be accurate and precise in your further requests, so that authors can understand and reply properly and focused.

Authors: please, check if there are further clarifications needed by the Reviewers.
Please, be focused in your final answers and avoid to ask questions to Reviewers, if not absolutely necessary.

For All: please, I would avoid a long chat-like discussion, a couple of iterations are affordable on a few specific points to be clarified, but no more.

Thanks and best regards

AC

---

### Decision · Program_Chairs · 2021-01-07
**Final Decision**

**Decision:**

Reject

**Comment:**

The paper deals with cross-domain few-shot learning in the case of large source-target domain shifts.

The paper received mostly below-threshold reviews, with one exception (R3) whose review is addressing more general aspects, but still with some concern, especially in relation to the experimental part (to which authors did not answer). R1's review is not of much help.

Clarity of the presentation and missing details seem to be recurrent issues all over the reviewers, together with remarks concerning the experimental validation, which would have required a deep revision and improvement, in particular regarding the use of more backbones, better ablation (Hebbian learner contribution, unclear initialization), processing times/computational complexity, significant comparative analysis re robust baselines.

The rebuttal clarifies some of the raised remarks but there are still issues, especially regarding Hebbian learning rule and ensemble learning strategies, and about results too, so not all reviewers were convinced to raise their ratings.

Overall, given the above issues, I consider the paper not yet ready for publication in ICLR 2021.

---

> ### Author Response · Authors · 2021-02-03
> **Response to final decision**
>
> We accept the area chair’s decision.
> We firmly believe that in this work we present a new approach to cross-domain few shot learning (depicting few-shot learning as domain shift) and a new idea (representation fusion) that leads to (strong) improvements in cross-domain few-shot learning tasks. We agree with the points made by the reviewers and will elaborate on the clarity of the paper and on better ablation, in detail: ablation studies on the influence of the different layers, comparison of Hebbian learners with other learners, discussion of the speed of our method, comparison to newest methods.